



# Novel method for determining $^{234}$U-$^{238}$U ages of Devils Hole 2 cave calcite

Xianglei Li[1], Kathleen A. Wendt[1,2], Yuri Dublyansky[2], Gina E. Moseley[2], Christoph Spötl[2], R. Lawrence Edwards[1]

[1]Department of Earth Sciences, University of Minnesota, 116 Church Street SE, Minneapolis 55455, USA
    [2]Institute of Geology, University of Innsbruck, Innrain 52, 6020 Innsbruck, Austria

*Correspondence to*: Xianglei Li (li000477@umn.edu)

**Abstract.** Uranium-uranium ($^{234}$U-$^{238}$U) dating can determine the age of secondary carbonates over greater time intervals than

the well-established $^{230}$Th-$^{234}$U dating method. Yet it is rarely applied due to unknowns surrounding the initial $\delta^{234}$U ($\delta^{234}$U$_i$) value, which result in significant age uncertainties. In order to understand the $\delta^{234}$U$_i$ in Devils Hole 2 cave, we have precisely determined 110 $\delta^{234}$U$_i$ values from phreatic calcite crusts using a $^{230}$Th-$^{234}$U chronology. The sampled calcite crusts were deposited in Devils Hole 2 between 4 and 590 thousand years, providing a long-term look at $\delta^{234}$U$_i$ variability over time. We then performed multi-linear regressions among the $\delta^{234}$U$_i$ values and correlative $\delta^{18}$O and $\delta^{13}$C values. These regressions allow

us to predict the $\delta^{234}$U$_i$ value of Devils Hole calcite based upon its $\delta^{18}$O and $\delta^{13}$C. Using this approach and measured present-day $\delta^{234}$U values, we calculate 110 independent $^{234}$U-$^{238}$U ages of Devils Hole 2 cave deposits. In addition, we used newly measured $\delta^{18}$O, $\delta^{13}$C, and present-day $\delta^{234}$U values to calculate 10 $^{234}$U-$^{238}$U ages that range between 676 and 731 thousand years, thus allowing us to extend the Devils Hole chronology beyond the $^{230}$Th-$^{234}$U-dated chronology while maintaining an age precision of ~2 %. Our results indicate that calcite deposition at Devils Hole 2 cave began no later than 736 ± 11 thousand

years ago. The novel method presented here may be used in future speleothem studies in similar hydrogeological settings, given appropriate calibration studies.

## 1 Introduction

The mid-20[th] Century discovery of $^{234}$U-$^{238}$U disequilibrium in natural waters (Cherdyntsev, 1955; Isabaev et al., 1960; Thurber, 1962) unlocked a new geochronometer for sediments and secondary deposits in marine and freshwater settings. The greatest

limitation of the $^{234}$U-$^{238}$U dating method, however, lies in the uncertainty of the initial $\delta^{234}$U at the time of deposition, as shown in Eq. 1:

$$\delta^{234}U_i = \delta^{234}U_p * e^{(\lambda_{234} * T)} \tag{1}$$



where $\delta^{234}U_p$ refers to the present $\delta^{234}U$ value, $\delta^{234}U_i$ refers to the $\delta^{234}U$ value at the time of deposition, $\lambda_{234}$ is the decay constant of $^{234}U$, and T refers to the time elapsed since deposition. In marine settings, $\delta^{234}U_i$ can be approximated using the

known excess $^{234}U$ activity in seawater. Ku (1965) was the first to test the $^{234}U$-$^{238}U$ geochronometer in marine sediments. Although this method has been applied successfully in marine-sourced secondary carbonates (Veeh, 1966; Bender et al., 1979; Ludwig et al., 1991), it has largely been limited as it is now clear that uranium can be mobile subsequent to deposition, such as in mollusks (Kaufman et al., 1971) and corals (Bender et al., 1979; Gallup et al., 1994).

In contrast to seawater, surface and ground waters exhibit a wide spatial and temporal variability in $\delta^{234}U$. As a result,

constraining the $\delta^{234}U_i$ of freshwater-sourced secondary carbonates is difficult. Uncertainties in past $\delta^{234}U_i$ result in $^{234}U$-$^{238}U$ age uncertainties that are orders of magnitude greater than those common for $^{230}Th$-$^{234}U$ dating technique. $^{230}Th$-$^{234}U$ dating has thus remained the preferred method for determining the age of secondary carbonates that have been deposited between modern day and 600 thousand years (ka) before present (BP), when $^{230}Th$ and $^{234}U$ are very close to secular equilibrium. However, with a firm understanding of past source water $\delta^{234}U_i$, the $^{234}U$-$^{238}U$ method represents a powerful geochronometer

that can reach much deeper in time (Gahleb et al., 2019).

Devils Hole (DH) and neighboring Devils Hole 2 (DH2) caves are ideal settings for the study of groundwater $\delta^{234}U_i$ variations over time. The walls of both steep fractures are coated with thick (up to ~90 cm) layers of calcite deposits that have precipitated subaqueously at a rate of approximately 1 mm per 1000 years (Ludwig et al., 1992; Moseley et al., 2016). Small variations in calcite $\delta^{234}U_i$ (1851-1616 ‰) over the last 500 ka BP have been precisely determined (Ludwig et al., 1992; Wendt et al., 2019).

Due to the high initial $^{234}U$ excess in DH and DH2 calcite (mean $\delta^{234}U_i$ = 1750 ‰), the $^{234}U$-$^{238}U$ method can be used to determine the age of cave calcite as old as 2.5 million years.

Ludwig et al. (1992) were the first to calculate $^{234}U$-$^{238}U$ ages from DH calcite. To do so, they derived the $\delta^{234}U_i$ value from 21 $^{230}Th$-$^{234}U$ ages between 60-350 ka. From this dataset they calculated the median $\delta^{234}U_i$ value (1750 ‰) and associated uncertainty (± 100 ‰); the latter was derived from the range of $\delta^{234}U_i$ over the selected time period. Using this $\delta^{234}U_i$ value,

18 $^{234}U$-$^{238}U$ ages were calculated. The ages ranged between 385-568 ka BP with mean uncertainties of 20 ka (3-5 % relative uncertainty) (Ludwig et al., 1992).

Building upon the pioneering work of Ludwig et al. (1992), we aim to decrease the uncertainties of DH and DH2 $\delta^{234}U_i$ in order to improve the precision of $^{234}U$-$^{238}U$ ages. Previous work in DH cave revealed a negative correlation between $\delta^{234}U_i$ and $\delta^{18}O$, and a positive correlation between $\delta^{234}U_i$ and $\delta^{13}C$ over the last 500 ka BP (Ludwig et al., 1992). Similar correlative

patterns were obtained over the last 200 ka BP using independently obtained drill cores from DH2 (Moseley et al., 2016). In this study, we show that by accounting for changes in $\delta^{234}U_i$ with respects to $\delta^{18}O$ and $\delta^{13}C$, we can reduce the uncertainty in





$\delta^{234}U_i$ by 35 %, thereby reducing the uncertainty in $^{234}U$-$^{238}U$ ages to about ± 13 ka within a several hundred-thousand-year range. We then use this method to calculate the age of calcite that was deposited in DH2 prior to (older than) the limit of $^{230}$Th-$^{234}$U dating (~600 ka BP). Doing so allows us to extend the independently derived radiometric chronology and determine the

time at which calcite first deposited in DH2.

## 1.2 Regional Setting

DH and DH2 caves are located 100 m apart in a detached area of Death Valley National Park in southwest Nevada (36°25′ N, 116°17′ W; 719 m above sea level). Bedrock of the study area is composed of carbonates from the Bonanza King Formation of middle and late Cambrian age (Barnes and Palmer, 1961). The caves follow a pair of deep, planar, steeply dipping fault-

controlled open fissures roughly 5 m wide, 15 m long, and at least 130 m deep (Riggs et al., 1994). Evidence for the tectonic origin of these caves includes the spreading and the orientation of their planar opening, which is perpendicular to the northwest-southeast principal stress direction that has prevailed in this part of the Great Basin for the last 5 million years (Carr, 1974).

DH and DH2 both intersect the water table of the Ash Meadows Groundwater Flow System (AMGFS), which is a large (~12,000 km$^2$) aquifer hosted in Paleozoic limestones (Winograd and Thordarson, 1975). The AMGFS is primarily recharged

by infiltration of snowmelt and rainfall in the upper elevations of the Spring Mountains (~500 mm a$^{-1}$; Winograd and Thordarson, 1975; Thomas et al., 1996; Winograd et al., 1998; Davisson et al., 1999). Quaternary extensional tectonics in this area has produced an underground network of open fractures which contribute to the high transmissivity of the aquifer. Previous studies suggest groundwater transit times of < 2000 years from the Spring Mountains to DH/DH2 caves (Winograd et al., 2006). Due to the long flow path (> 60 km), prolonged residence time, and the retrograde solubility of calcite, the

groundwater flowing southwest through both caves is very slightly supersaturated with respect to calcite (SI = 0.2; Plummer et al., 2000). The caves are < 1.5 km upgradient from a line of springs that represent the primary discharge area of the AMGFS.

Calcite has been continuously depositing as dense mammillary crusts on the submerged walls DH and DH2 over much of the last 1 million years at a very slow rate of roughly 1 mm ka$^{-1}$ (Ludwig et al., 1992; Winograd et al., 2006; Moseley et al., 2016). The thickness (≤ 90 cm) of mammillary calcite crusts implies a long history of calcite-supersaturated groundwater. Regional

groundwater transmissivity is maintained despite calcite precipitation due to active extensional tectonics (Riggs et al., 1994).

## 2 Methods

A 670 mm-long core was drilled from the hanging wall of DH2 cave at + 1.8 m relative to the modern water table (r.m.w.t.). The first 654 mm of the core consists of calcite; the last 16 mm of the core consists of bedrock. The core consists of two types of calcite: folia and mammillary calcite. For a full description of the petrographic and morphological differences between both

forms of calcite see Wendt et al. (2018). Briefly, mammillary calcite precipitates subaqueously, while folia calcite forms at the water table in a shelf-like formation. The presence of folia in the core is an indicator of paleo-water table near + 1.8 m r.m.w.t.





The selected core was cut longitudinally and polished. The core was surveyed for growth hiatuses and features indicative of changing deposition mechanisms and rates (such as folia). Folia was identified at 77.7-97.4 mm (as reported in Moseley et al., 2016), 171.4-199.2 mm, 209.4-229.0 mm, and 305.0-323.0 mm (distances are reported from top of the calcite sequence). In
addition, a growth hiatus was discovered between 587.4 and 589.0 mm (supplementary Fig. 1).

The mammillary calcite portions of the core were $^{230}$Th-$^{234}$U dated at regular intervals (n = 110). As described in Moseley et al. (2016), folia calcite cannot be reliably dated. Results for the first 91 $^{230}$Th-$^{234}$U ages were published by Moseley et al. (2016) and Wendt et al. (2019). Nine additional $^{230}$Th-$^{234}$U ages were measured between 351.0 and 562.0 mm using identical methodology to the aforementioned publications. The purpose of additional $^{230}$Th-$^{234}$U ages is to extend the DH2 chronology
toward secular equilibrium (about 600 ka BP). Between 608.2-652.0 mm, 10 new $\delta^{234}$U measurements were collected for this study. To do so, calcite powders were hand drilled at approximately 1 cm intervals and spiked following the $^{230}$Th-$^{234}$U methods cited above. The uranium aliquots were then extracted and measured following the methods described in Cheng et al. (2013). Chemical blanks were measured with each set of 10–15 samples and were found to be negligible (< 50 ag for $^{230}$Th, < 100 ag for $^{234}$U, and < 1 pg for $^{232}$Th and $^{238}$U).

Samples for stable isotope measurements were micromilled continuously at 0.1-0.2 mm intervals along the core axis between 0-158 mm and presented by Moseley et al. (2016). The values of two to three stable isotope measurements (0.1-0.2 mm in width) were averaged in order to pair with $^{230}$Th-$^{234}$U subsamples, which averaged 0.3 to 0.5 mm in width. Between 169.8-652.0 mm, 66 new stable isotope samples were micromilled at the location of each $^{230}$Th-$^{234}$U and $\delta^{234}$U measurement published by Wendt et al. (2019). Similarly, 2-3 stable isotope measurements were averaged to encompass the width of uranium isotope
subsamples. Calcite powders were analyzed using a Delta V plus isotope ratio mass spectrometer interfaced with a Gasbench II. Values are reported relative to VPDB with 1-sigma precisions of 0.06 and 0.08‰ for $\delta^{13}$C and $\delta^{18}$O, respectively.

A statistical model was built to predict $\delta^{234}$U$_i$ values using $\delta^{18}$O and $\delta^{13}$C based on the correlation observed between measured $\delta^{234}$U$_i$ and stable isotope values from 4 to 590 ka BP. Software OriginPro (version 2015) was used to conduct all correlation and regression analyses (Moberly et al., 2018). Several types of models fitted with linear, quadratic, and cubic regression
methods were built. The model that provided the best estimate of the measured $\delta^{234}$U$_i$ values was selected. Using this model, a statistically derived (SD) $\delta^{234}$U$_i$ value can be determined based on known $\delta^{18}$O and $\delta^{13}$C values. The SD $\delta^{234}$U$_i$ and measured $\delta^{234}$U$_p$ can then be used to calculate $^{234}$U-$^{238}$U ages (Eq. 1). The $^{234}$U decay constant of $2.82206 \pm 0.00302$ $10^{-6}$ a$^{-1}$ (Cheng et al., 2013) was used. We validated our methodology and uncertainty estimates by comparing $^{230}$Th-$^{234}$U and $^{234}$U-$^{238}$U dates obtained for samples younger than 590 ka BP.

Using this dataset, we calculated 120 $^{234}$U-$^{238}$U ages for samples in total. With these $^{234}$U-$^{238}$U ages as input, we calculated an age model using the Bayesian statistical software OxCal version 4.2 (Bronk Ramsey and Lee, 2013). Age models were





calculated under deposition sequence "P" with k-parameter set to 0.1 (Bronk Ramsey and Lee, 2013). The positions of growth hiatuses, including folia calcite, were incorporated into the age model as growth boundaries.

## 3 Results

The new $^{230}$Th-$^{234}$U ages (denoted in subsequent text as $^{230}$Th ages for simplicity) are in stratigraphic order within uncertainties. $^{238}$U and $^{232}$Th concentrations fall within the range of previous data published by Moseley et al. (2016) and Wendt et al. (2019). The time-depth consistency and reproducibility of ages argue against open-system processes. The existence of a ca. 67-ka-long growth hiatus between 587.4-589.0 mm is supported by U-series ages (see supplementary materials).

In this study we split the $\delta^{234}U_i$ dataset into three sections (4-309 ka, 309-355 ka and 355-590 ka BP). The sections were
divided according to the level of uncertainty in $\delta^{234}U_i$ derived from $^{230}$Th ages (see Table 1 and Supplementary Table 2). The average $\delta^{234}U_i$ was the same within uncertainties regardless of how we grouped the data, implying no detectable trend in $\delta^{234}U_i$ with time.

**Table 1: Statistics of $\delta^{234}U_i$ with different temporal groupings.**

| Time range in ka BP (# of samples) | Average $\delta^{234}U_i$ (‰) | Standard deviation of population (‰, 2σ) | Average precision (‰, range) |
|---|---|---|---|
| 4-309 (66) | 1761 | 99 | 5 (2-14) |
| 309-355 (15) | 1758 | 71 | 22 (13-33) |
| 355-590 (29) | 1787 | 120 | 50 (15-162) |

DH2 oxygen isotope values reveal a negative correlation with $\delta^{234}U_i$ over the last 590 ka BP (0-578 mm along the core axis; r
= -0.52 (n = 110, $p \le 0.05$)), whereas carbon isotopes reveal a positive correlation (r = 0.71 (n = 110, $p \le 0.05$); Fig. 1 and Table 2). The linear relationships presented here are consistent with results from Moseley et al. (2016) and Ludwig et al. (1992) in DH2 and DH cave, respectively. The anti-correlation between $\delta^{18}O$ and $\delta^{234}U_i$ is consistent with the interpretation presented in Wendt et al. (2019) such that periods of increased regional moisture availability (due to cooler, wetter conditions favoring depleted $\delta^{18}O$ values) are associated with increased DH $\delta^{234}U_i$ values. A full list of correlation calculations among $\delta^{234}U_i$, $\delta^{18}O$
and $\delta^{13}C$ is presented in Table 2. The linear relationship between $\delta^{13}C$ and $\delta^{234}U_i$ is closer than that between $\delta^{18}O$ and $\delta^{234}U_i$ for the same period. This is likely due to the fact that $\delta^{13}C$ and $\delta^{234}U_i$ are forced by local processes, including changes in vegetation density at the principle recharge zone and groundwater interaction with bedrock in the vadose zone (see Coplen et al., 1994 and Wendt et al. 2019 for proxy interpretation). Since changes in the local environmental and hydrological regime are closely interconnected, we expect similar trends in the timing and pattern of $\delta^{13}C$ and $\delta^{234}U_i$ signals. In contrast, DH/DH2
$\delta^{18}O$ reflects the $\delta^{18}O$ of meteoric precipitation, which is sensitive to atmospheric temperatures and changes in moisture source





(Winograd et al., 1998; Mosley et al., 2016). Evidence suggests no secular temperature changes in the local aquifer over the last several glacial-interglacial cycles (Kluge et al., 2014; J. Fiebig, pers. comm.), thus DH/DH2 $\delta^{18}$O is expected to be minimally influenced by aquifer-related processes. Overall, we expect a greater scatter in the $\delta^{18}$O vs. $\delta^{234}$U$_i$ regression due to the compounding forcings that influence $\delta^{18}$O on a much larger spatial scale.

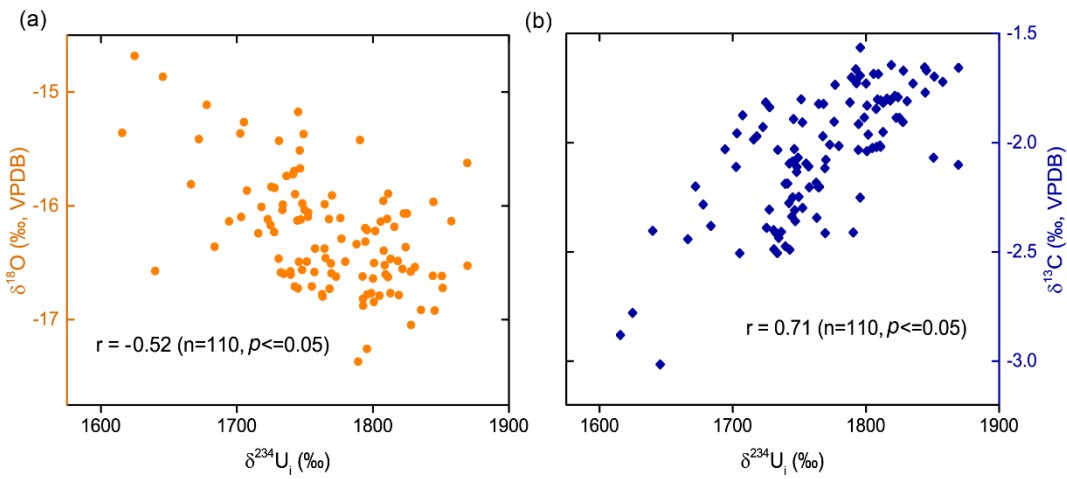


**Figure 1: Correlation between DH2 $\delta^{18}$O, $\delta^{13}$C, and $^{230}$Th-derived $\delta^{234}$U$_i$ over the last 590 ka.**

**Table 2: Correlation coefficient (r) for pairs of $\delta^{18}$O, $\delta^{13}$C, and $\delta^{234}$U$_i$.**

| Time range in ka BP (# of samples) | Correlation coefficient (r) | | |
|---|---|---|---|
| | $\delta^{234}$U$_i$ - $\delta^{18}$O | $\delta^{234}$U$_i$ - $\delta^{13}$C | $\delta^{18}$O - $\delta^{13}$C |
| 4-590 (110) | -0.524 | 0.709 | -0.543 |
| 4-355 (81) | -0.640 | 0.691 | -0.610 |
| 4-309 (66) | -0.675 | 0.718 | -0.557 |

**Note: all the r values are statistically significant ($p \leq 0.05$).**

From Table 2, we can see that the modulus value $\lvert r \rvert$ for $\delta^{18}$O and $\delta^{234}$U$_i$ is greatest in the 4 to 309 ka BP dataset, which

represents the time span with the highest precision $\delta^{234}$U$_i$ values. In contrast, the $\lvert r \rvert$ values for $\delta^{13}$C and $\delta^{234}$U$_i$ are similar regardless of $\delta^{234}$U$_i$ precision. We therefore explored various time ranges and types of regressions in order to determine the most precise predictor of $\delta^{234}$U$_i$ based upon $\delta^{18}$O and/or $\delta^{13}$C.

### 3.1 Regression Analysis

We evaluated the linear and polynomial (quadratic and cubic) regression methods primarily by calculating the coefficient of

determination (COD), i.e. $R^2$. The $R^2$ value represents the percentage of variation of the SD $\delta^{234}$U$_i$ in terms of the total of





observed $\delta^{234}U_i$. To further compare the robustness between different models, we adjust the $R^2$ values from the different numbers of predictors, i.e. the degree of freedom (DF) of the predictors. The adjusted $R^2$ (Adj. $R^2$) values is shown in Table 3.

**Table 3: Adjusted $R^2$ values for different models with various regression analysis methods.**

| Time period in ka BP | $\delta^{18}O$ | $\delta^{13}C$ | Both |
|---|---|---|---|
| Linear Regression (Adj. $R^2$) | | | |
| 4-590 | 0.23 | 0.50 | 0.52 |
| 4-355 | 0.41 | 0.48 | 0.54 |
| 4-309 | 0.46 | 0.52 | **0.63** |
| Quadratic Regression (Adj. $R^2$) | | | |
| 4-590 | 0.28 | 0.50 | 0.52 |
| 4-355 | 0.43 | 0.49 | 0.55 |
| 4-309 | 0.46 | 0.52 | 0.60 |
| Cubic Regression (Adj. $R^2$) | | | |
| 4-590 | 0.38 | 0.50 | 0.51 |
| 4-355 | 0.42 | 0.48 | 0.54 |
| 4-309 | 0.47 | 0.52 | 0.61 |

**Note: the robustness of the regression model can be evaluated by coefficient of determination (COD), $R^2$, which is defined as $R^2 = 1$ - (residual sum of squares, RSS)/(total sum of squares, TSS). Adj. $R^2$ = 1- (RSS/(DF of residual))/(TSS/(DF of total predictors)).**

Among the various models, the multiple linear regression (MLR) for the time period of 4-309 ka BP in terms of both $\delta^{18}O$ and $\delta^{13}C$ yielded the highest Adj. $R^2$ value of 0.63, such that over this time span the model accounts for 63 % of the $\delta^{234}U_i$ variability (Table 3). The corresponding equation is as follows (cf. Table 4):

$$SD\ \delta^{234}U_i = 1.2 \times 10^3 - 44 * \delta^{18}O + 86 * \delta^{13}C \tag{2}$$

We then applied statistical methods to test the robustness of the chosen model, starting with the F test. This tests whether the model differs significantly from a 'y = constant' model. The F value is computed by dividing the mean square of the fitted model by the mean square of the residual. The more this ratio deviates from 1, the stronger the indication that the model differs from the 'y = constant' model. The test returned F = 52.5, far larger than the critical value of the F test at a significance level of $\alpha = 0.01$ (DF of numerator = 2; DF of denominator = 65) ($F_{crit}$ = 4.95), indicating that the model is tenable.

**Table 4: T test table of the multiple linear regression model for 4-309 ka BP.**

| Parameters | Value | Standard Error | t value | *p* value |
|---|---|---|---|---|
| Intercept | 1.2E3 | 1.9E2 | 6.5 | 1.8E-08 |
| Factor of $\delta^{18}O$ | -44 | 10 | -4.3 | 6.3E-05 |
| Factor of $\delta^{13}C$ | 86 | 16 | 5.3 | 1.4E-06 |





**Note: a smaller *p* value represents a decreased likelihood that the parameter is equal to zero.**

Second, we performed a "t-test" to check if every term in the MLR for 4-309 ka is significant. The t value is the ratio of the fitted value to its standard error. As shown in the Table 4, all the fitted values (coefficients and intercept) are significant at a significance level of $\alpha = 0.001$. Thus, the regression model is robust.

## 3.2 Residual Analysis

We now estimate the uncertainty of the SD $\delta^{234}U_i$ values by analyzing the residuals, defined as the differences between the observed and predicted values. Figure 2 reveals that the residuals are basically normally distributed. Thus, we conclude that the regression model captures the dominant characteristic of variability of the observations. The estimate of the residuals yields uncertainty of $\pm 60.5$ ‰ (95 % confidence interval) with an average of essentially zero (4.4E-13). We take the $\pm 60.5$ ‰ value to represent a constant uncertainty for all the SD $\delta^{234}U_i$ values, amounting to a ~40 % reduction in $\delta^{234}U_i$ uncertainties (the original estimate of uncertainty from Ludwig et al. (1992) was 100 ‰). Figure 3 plots residuals versus time. The residuals have a full range of about 120 ‰ and exhibit multi-millennial to orbital scale variability.

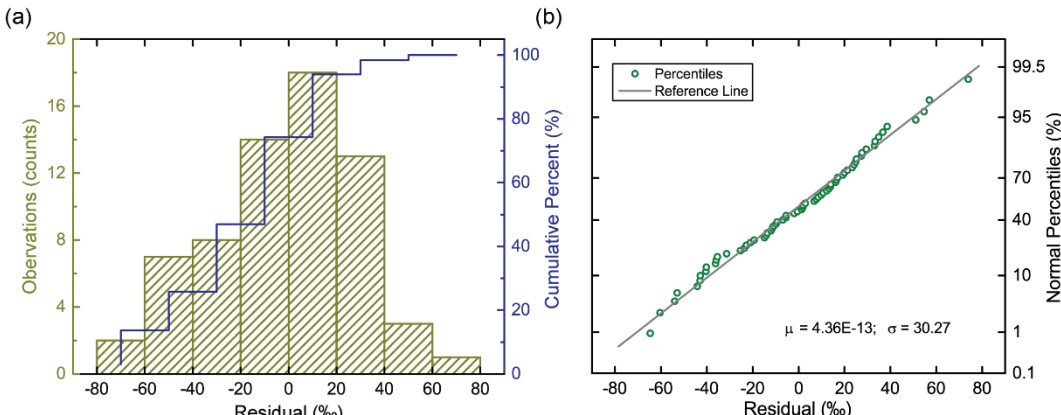

**Figure 2: Histogram (left) and normal percentiles plot (right) of the residual. The reference line in the right plot refers to the line of standard normal distribution.**

## 3.3 $^{234}U$-$^{238}U$ Ages

Using Eq. 1 and 2, we can calculate the $^{234}U$-$^{238}U$ ages (denoted in subsequent text $^{234}U$ for simplicity) for each data point for which $\delta^{234}U_p$, $\delta^{18}O$ and $\delta^{13}C$ are measured. The final uncertainty of $^{234}U$ ages comes from two sources: 1) the uncertainty of the model and 2) the uncertainty in determination of $\delta^{234}U_p$. Combined, the final uncertainty of $^{234}U$ ages before 590 ka BP is approximately $\pm 13$ ka ($2\sigma$), which represents a 35 % improvement relative to previously reported $^{234}U$ ages from DH ($\pm 20$ ka; Ludwig et al., 1992).




The $^{234}$U and $^{230}$Th ages between 4 to 590 ka BP are consistent within uncertainties (Fig. 4), with the exception of 4 ages (out of 110) at 27 ka, 348 ka, 410 ka and 503 ka BP). Since uncertainties are reported at the 95 % confidence level, we would expect this number of statistical outliers and conclude that our analysis is overall internally consistent. Alternately, there may

be some unknown underlying process not captured by our analysis, which may be consistent with the variability of the residual (Fig. 3).

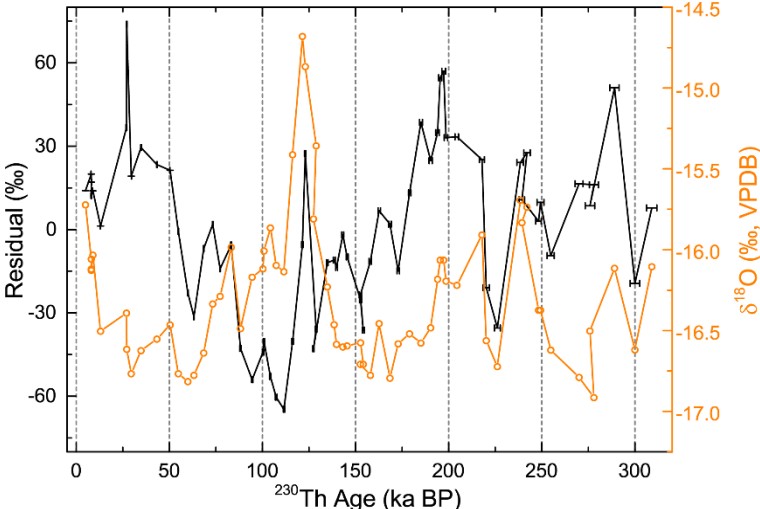

**Figure 3: Variability of the residual and δ$^{18}$O versus $^{230}$Th age between 4 and 309 ka BP. The 2σ error bars for $^{230}$Th ages are shown on the points of residual.**

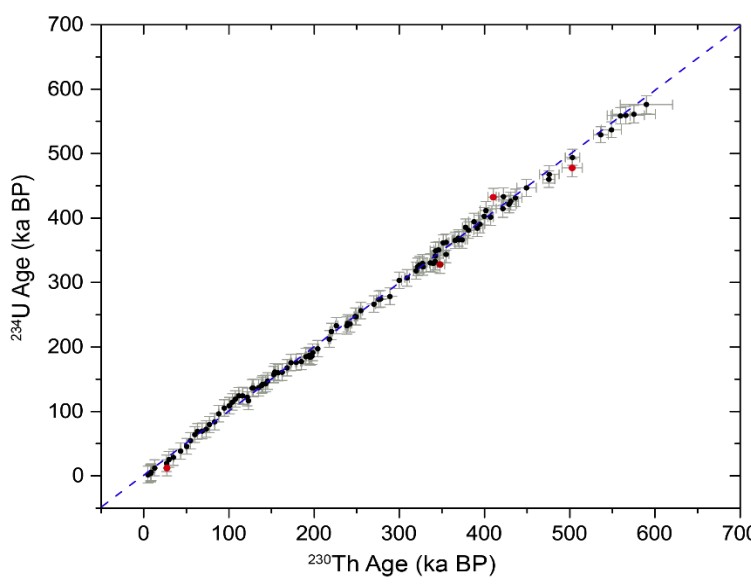


**Figure 4: plot in the $^{234}$U ages against the $^{230}$Th ages between 4 to 590 ka BP and the 2σ uncertainty is shown in grey error bar. The dashed line is 1:1 line and the red dots refer to the points inconsistent between the two kinds of ages.**





### 3.4 $^{234}$U ages beyond $^{230}$Th–$^{234}$U secular equilibrium

The consistency between $^{234}$U and $^{230}$Th ages within the time range 4 to 590 ka BP suggests that it is reasonable to utilize this

model to calculate ages that are close to or beyond $^{230}$Th-$^{234}$U secular equilibrium. Using measured $\delta^{234}U_p$, $\delta^{18}$O and $\delta^{13}$C values, we calculated 10 $^{234}$U ages for DH2 cave deposits older than 600 ka BP over the depth of 608 to 652.0 mm (Table 5). The $^{234}$U ages are in stratigraphic order within uncertainty.

Table 5: SD $\delta^{234}U_i$ and calculated $^{234}$U ages from calcite deposited prior to 600 ka BP and the corresponding depth, $\delta^{18}$O, $\delta^{13}$C and $\delta^{234}U_p$ values.

| Depth | $\delta^{18}$O | $\delta^{13}$C | $\delta^{234}U_p$ | SD $\delta^{234}U_i$ | $^{234}$U Age |
|---|---|---|---|---|---|
| (mm) | (‰, VPDB) | (‰, VPDB) | (‰)(2σ) | (‰)(2σ) | (a BP)(2σ) |
| 608.20 | -16.22 | -1.91 | 262.7±1.7 | 1773.3±60.5 | 676,400±14,400 |
| 611.80 | -16.19 | -1.81 | 262.1±1.9 | 1780.1±60.5 | 678,600±14,700 |
| 618.20 | -15.96 | -1.90 | 257.7±2.2 | 1762.2±60.5 | 681,100±15,200 |
| 623.40 | -15.66 | -2.00 | 247.6±1.6 | 1740.4±60.5 | 690,700±14,600 |
| 625.60 | -15.53 | -2.17 | 245.6±2.0 | 1719.9±60.5 | 689,500±15,400 |
| 630.00 | -15.60 | -1.99 | 241.0±1.5 | 1738.4±60.5 | 699,900±14,600 |
| 634.00 | -16.29 | -1.95 | 232.7±1.7 | 1773.1±60.5 | 719,300±14,700 |
| 638.00 | -16.74 | -1.87 | 229.4±1.5 | 1799.0±60.5 | 729,600±14,200 |
| 639.80 | -16.13 | -1.78 | 228.9±1.9 | 1780.0±60.5 | 726,500±14,900 |
| 652.00 | -16.12 | -1.92 | 229.4±1.8 | 1767.8±60.5 | 723,300±14,900 |

## 4 Discussion

### 4.1 Uncertainties of $^{234}$U ages

Due to the high initial $\delta^{234}$U values (1760 ± 61 ‰), DH/DH2 calcite can theoretically be dated up to 2.5 million years (Ma) BP, assuming that measured $\delta^{234}U_p$ values have about 1 ‰ uncertainty. By including all the uncertainty above, we could obtain the following age uncertainty (2σ): ~13 ka uncertainty for a sample < 500 ka BP, 16 ka at 1.0 Ma BP, 26 ka at 1.5 Ma BP, 70

ka at 2.0 Ma BP and 290 ka at 2.5 Ma BP. A rapid increase in uncertainty after 2.5 Ma BP (Fig. 5) is due to the fact that ≥ 10 half-lives of the $^{234}$U will have elapsed and the $\delta^{234}U_p$ values increasingly approach the uncertainty of their measurement.

### 4.2 Timing and rate of calcite deposition

Figure 6 shows an OxCal-derived age model with 95 % confidence intervals plotted over depth using all $^{234}$U ages with their 2σ uncertainties. Location of folia and a growth hiatus are highlighted. Two lines of evidence suggest that the resulting time

series is reasonable. Firstly, the average growth rate during the whole period is at 0.9 ± 0.3 mm ka$^{-1}$ (1σ uncertainty; Fig. 7),




which is broadly consistent with growth rates reported by Moseley et al. (2016). Secondly, all $^{234}$U ages are in stratigraphic order within uncertainties and fall within the 95 % confidence interval of our age model, implying no major outliers.

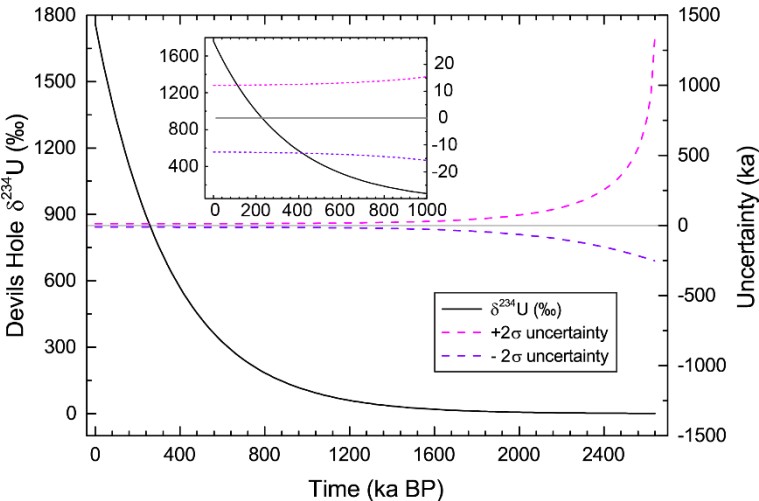

**Figure 5: Evolution of $\delta^{234}$U and corresponding uncertainties of $^{234}$U ages with time. The insert plot shows $\delta^{234}$U evolution during the period of 0-1 Ma.**

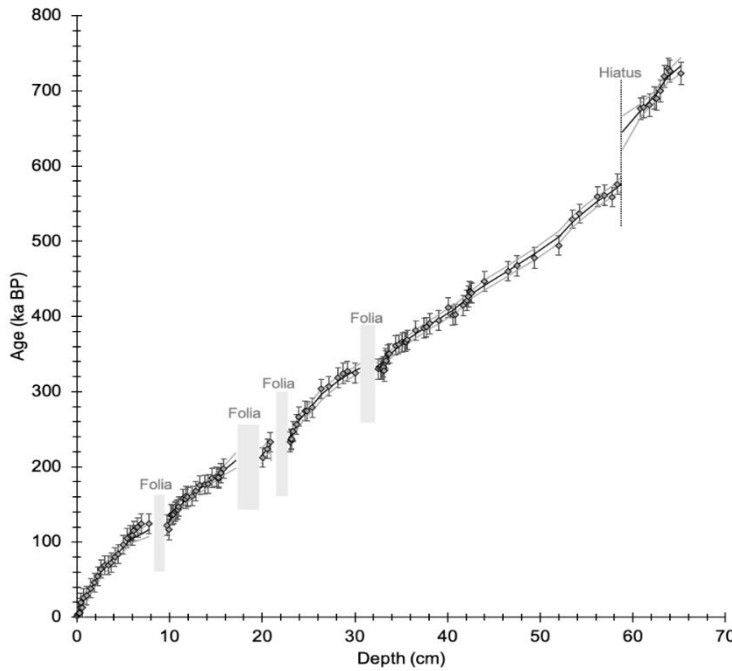

**Figure 6: $^{234}$U ages (diamonds) and associated 2σ uncertainties plotted over the calcite crust. OxCal age model (black line) and 95 % confidence limits (grey lines) was derived from $^{234}$U ages. Location of folia (grey rectangles) and a growth hiatus are indicated.**





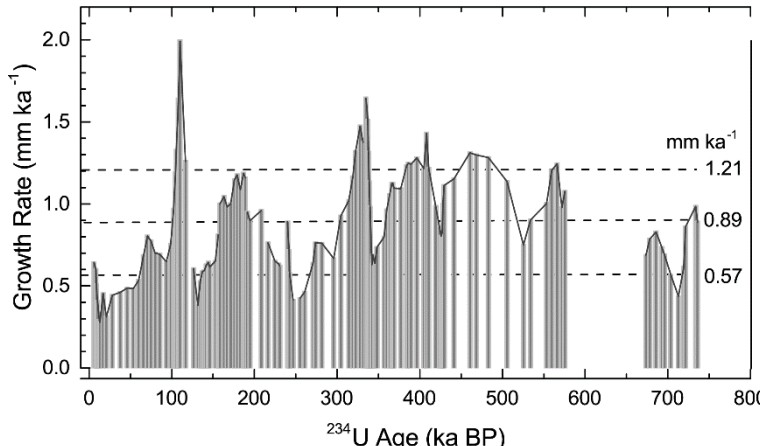


**Figure 7: Growth rate of DH2 mammillary calcite based on the OxCal age model (cf. Fig. 6). The dashed lines indicate the average value and one standard deviation above and below the average.**

Previous investigations revealed that DH2 cave opened to the surface at approximately 4 ka BP (Moseley et al., 2016), likely due to surface collapse processes (Riggs et al., 1994). The timing at which the main fissure opened, however, remains largely

unknown. By $^{234}$U dating the oldest calcite deposited on our studied core (at the calcite-bedrock boundary), we can determine the earliest-possible timing at which the DH2 fissure existed (without which calcite cannot precipitate). $^{234}$U ages in the last 170 mm of the core are in stratigraphic order within uncertainties (Fig. 8) yet due to the slight scatter of absolute ages, we utilize our OxCal age model to extrapolate the age at calcite-bedrock boundary. In doing so, we determine the earliest calcite deposition at our study location (+ 1.8 m) to 736 ± 11 ka BP (Fig. 8).

The onset of calcite deposition in DH2 cave is in agreement with the recent geologic history of this region. The orientation of DH and DH2 are in accord with the principal northwest-southeast stress direction in this part of the Great Basin that has prevailed over the last 5 Ma (Carr, 1974), suggesting that one or both fissures formed after 5 Ma BP. Abundant calcareous and siliceous spring and marsh deposits in Ash Meadows and the Amargosa Desert of Pliocene age (2.1 to 3.2 Ma BP; Hay et al., 1986) and groundwater-deposited calcite veins in alluvium and colluvium of Pleistocene age (500 to 900 ka BP; Winograd

and Szabo, 1988) indicate that groundwater in the discharge zone of AMGFS has been continuously supersaturated with respect to calcite for at least the last 3 Ma. Our results, which suggest that the DH2 fissure opened no later than 736 ka BP, is therefore in agreement with the modern understanding of the AMGFS' geological history.



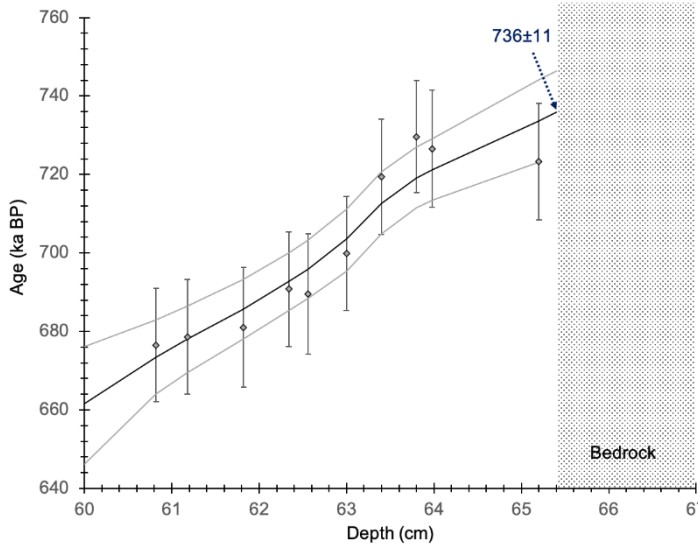

**Figure 8: OxCal-derived age model (black line), 95 % confidence limits (grey lines), $^{234}$U ages (diamonds) with 2σ uncertainties for the last 17 cm of the core. The time series indicates that the initiation of calcite growth at the base of the core at 736 ± 11 ka BP.**

## 5 Conclusions

We have developed a novel method to determine past $\delta^{234}U_i$ values in DH/DH2 calcite. We established a multiple linear regression between $\delta^{234}U_i$, $\delta^{18}O$, and $\delta^{13}C$ values over the past 590 ka. The $\delta^{234}U_i$ of calcite older than 600 ka BP, which is

beyond the limits of $^{230}$Th- $^{234}$U dating, can be determined using this regression. Uncertainty associated with SD $\delta^{234}U_i$ is ± 61 ‰, thereby improving precision by 40%. Using this technique, we have calculated 10 new $^{234}$U-$^{238}$U ages in the oldest part (> 600 ka BP) of a DH2 calcite sample. New $^{234}$U-$^{238}$U ages range from 676 to 720 ka BP. Average relative $^{234}$U-$^{238}$U age uncertainties are 2%. The concordance between the $^{234}$U-$^{238}$U and the $^{230}$Th-$^{234}$U ages younger than 600 ka BP indicates that the DH2 calcite behaves as a closed system. Using our time series, we determined that calcite at the base of the studied core

was deposited at 736 ± 11 ka BP. We argue that this age marks the latest possible time at which the DH2 fissure existed.

## Author contribution

R.L.E. conceptualized the project and X.L. carried it out. K.A.W., X.L. and R.L.E. prepared the manuscript with contributions from all co-authors. X.L. did the formal analysis. C.S. and Y.D. provided the cave samples. K.A.W. and G.E.M. did the $^{230}$Th dating work and contributed to stable isotope measurements. All authors discussed the results and provided input to the

manuscript and technical aspects of the analyses.





**Acknowledgements**

This work was supported by the National Science Foundation project number 1602940 (to R.L.E.). This research was conducted under research permit numbers DEVA-2010-SCI-0004 andDEVA-2015-SCI-0006 issued by Death Valley National Park. We thank K. Wilson for assistance in the field and M. Wimmer for assistance in the laboratory.

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
