# Peer review of "Novel method for determining $^{234}\text{U}$ - $^{238}\text{U}$ ages of Devils Hole 2 cave calcite"

_Geochronology, 2020_

## Referee Comment (RC1) · Anonymous Referee #1 · 15 Sep 2020

General Comments: This is an interesting and excellent study that uses the correlation observed between the cave calcite $\delta$234Ui and stable isotopes in the core from Devis Hole 2 to establish a multilinear model for prediction of the $\delta$234Ui. This model allows the authors to predict much precisely a value of $\delta$234Ui and thus to improve the precision of 234U-238U dating method. Major Comments: 1) It lacks a figure in the MS to show the $\delta$234Ui, $\delta$18O and $\delta$13C time-series. 2) For the regression analysis, the authors split the observed $\delta$234Ui into three groups according to the age range and the precision. However, it will be better to find a weighting method, which can help to take into account all of the observed values of $\delta$234Ui over the past 590 ka. 3) The figure 3 shows the variability of the residual and $\delta$18O versus 230Th age, but its significance was poorly explained. Minor Comments: 1) It will be better to add some

information about the relationship between U-concentration and $\delta$234Ui in the results. 2) It is difficult to identify the difference in precision between the 234U ages and the 230Th ages from the figure 4. It will be better to provide some detailed comparisons of the two dating ages in terms of precision. 3) In the conclusions, the authors should acknowledge that the conclusions are based on the regression analysis of the $\delta$234Ui, $\delta$18O and $\delta$13C datasets over the past 309 ka, but not over the past 590 ka.

---

## Referee Comment (RC2) · Anonymous Referee #2 · 22 Sep 2020

Li et al. present a method to extend the 230Th-234U-dated chronology of Devils Hole 2 calcite. The approach uses multi-linear regressions between calcite $\delta$234U , $\delta$28O and $\delta$13C values to predict initial $\delta$234U variability, which allows to calculate 234U-238U-ages until 731ka BP with an average age precision of about 2%.

**General comments:**

Based on sound analytical methods the authors present an impressive data set and an innovative approach which definitely deserves publication in this journal. However, prior acceptance, I would suggest that the authors clarify some of their statistical methods and the estimation of their uncertainties. One major point is that most paleo climate time series are impacted by autocorrelation (e.g. Macias-Fauria et al. (2012);Hu et al. (2017), and others). Serial correlation is known to reduce the degrees of freedom

of the time series and has to be taken into account, by, e.g. adjusting the p-value or estimating appropriate confidence intervals (Olafsdottir and Mudelsee, 2014; Zwiers and von Storch, 1995;Mudelsee, 2003). I strongly recommend that the authors discuss this issue, such as to which extent their data is affected by autocorrelation, and how this influences their results. I would further recommend to show some evidence that the derived correlation and the regression model are not dependent on single values (such as the few data points with the lowest $\delta$234U in Fig. 1) and/or the choice of the calibration interval.

**Minor comments:**

L108ff: It is unclear which statistical method and/or settings of OriginPro have been used and how the analytical uncertainties are propagated to the predicted $\delta$234U values. OriginPro does not automatically include the uncertainties of both the y and x values in correlation and regression analyses. However, it also allows to calculate confidence as well as prediction intervals. So please clarify. . .

L132-134: In my opinion, the manuscript would be of even more value for the broader scientific community, if the main points of the proxy interpretation from the Devils Hole calcite deposits would be summarized in 1-2 more sentences. In my opinion, the description of the mechanistic understanding of the underlying processes is too short, and the authors focus mainly on the statistics. I understand that this is not the scope of the manuscript, but to support the statistical model, a proper mechanistic understanding of the underlying processes is essential. In the current version, however, one is referred to the numerous previous DH publications, which are probably not familiar to potential readers.

L136-138: See previous comment. A bit more explanation of the processes would be very helpful.

L145/Figure 1: Please visualize the applied linear regression model and its uncertainties

L148: Compare previous comments, please state if the r and p values are corrected for autocorrelation

L156-157: Again, does the adjustment of $R^2$ already take autocorrelation into account?

L168: What is the critical value of the F-test when adjusting the DF for an auto-correlated time series?

L175-176: Which part of the data is treated in this part? The whole 590ka interval? Please clarify which values are used here for the validation of the regression model.

L178-179: According to Table 4, the standard errors of the model coefficients are in the order of 15-20%, so how does the estimate of the residual uncertainty stated here compare to the uncertainty of the regression model itself? Does the width of the histogram change when taking into account the uncertainty of the regression model?

L194-195: The variability of the residuals may originate in the method used for calibration. When calibrating using linear regression, the variance of the proxy time series is always less than that of the calibration data set, since the resulting amplitude reductions are dependent on the correlation between the proxy and the calibration data set (Esper et al., 2005).

L296: Reference not in alphabetical order.

Supplementary material:

L20: Derived

**References:**

Esper, J., Frank, D. C., Wilson, R. J., and Briffa, K. R.: Effect of scaling and regression on reconstructed temperature amplitude for the past millennium, Geophysical Research Letters, 32, 2005.

Hu, J., Emile-Geay, J., and Partin, J.: Correlation-based interpretations of paleoclimate

data – where statistics meet past climates, Earth and Planetary Science Letters, 459, 362-371, 10.1016/j.epsl.2016.11.048, 2017.

Macias-Fauria, M., Grinsted, A., Helama, S., and Holopainen, J.: Persistence matters: Estimation of the statistical significance of paleoclimatic reconstruction statistics from autocorrelated time series, Dendrochronologia, 30, 179-187, https://doi.org/10.1016/j.dendro.2011.08.003, 2012.

Mudelsee, M.: Estimating Pearson's correlation coefficient with bootstrap confidence interval from serially dependent time series, Mathematical Geology, 35, 651-665, Doi 10.1023/B:Matg.0000002982.52104.02, 2003.

Olafsdottir, K. B., and Mudelsee, M.: More accurate, calibrated bootstrap confidence intervals for estimating the correlation between two time series, Mathematical Geosciences, 46, 411-427, 10.1007/s11004-014-9523-4, 2014.

Zwiers, F. W., and von Storch, H.: Taking Serial-Correlation into Account in Tests of the Mean, Journal of Climate, 8, 336-351, Doi 10.1175/1520-0442(1995)008<0336:Tsciai>2.0.Co;2, 1995.
* * *

---

## Author Comment (AC1) · 3 Nov 2020

General Comments: This is an interesting and excellent study that uses the correlation observed between the cave calcite $\delta$234Ui and stable isotopes in the core from Devis Hole 2 to establish a multilinear model for prediction of the $\delta$234Ui. This model allows the authors to predict much precisely a value of $\delta$234Ui and thus to improve the precision of 234U-238U dating method.

We greatly appreciate the valuable comments from the reviewers of our work. We revised our manuscript, according to the reviewers' comments, questions, and suggestions. We believe that the manuscript has been further improved

Major Comments:

[Figure]

1) It lacks a figure in the MS to show the $\delta$234Ui, $\delta$18O and $\delta$13C time-series.

We added the plots in the figure 1 in the MS. Please see the attached Figure 1.

2) For the regression analysis, the authors split the observed $\delta$234Ui into three groups according to the age range and the precision. However, it will be better to find a weighting method, which can help to take into account all of the observed values of $\delta$234Ui over the past 590 ka.

We calculated the regression models with the instrumental weighting method (weighting=1/square of error) for $\delta$234Ui in terms of the three groups, and all the three models were closely consistent with each other. Thus, it is feasible to establish the model by using the dataset over the past 309 ka period, even better on account of less uncertainty of $\delta$234Ui. Then by comparing between the regression model used in the MS and the one with weighting method over the same period, we found both models are significantly consistent with a much higher linear correlation coefficient of r=0.98 (n=66, p<0.05). Furthermore, significant correlation of d234U (no weighting) with both d13C and d18O supported us to establish the model without weighting, which also benefited us to express the model in a simpler way. Also, the residual analysis showed that the model in the MS had a little smaller variance of residual, although the adjusted R2 value seems a little bit higher in the model with the weighting method.

Based on the discussion above, we will keep the model used in the MS but with more confidence. Please find the supplementary file attached about the models and residual analysis.

3) The figure 3 shows the variability of the residual and $\delta$18O versus 230Th age, but its significance was poorly explained.

Currently we have little knowledge about the variability of residual and it has very poor relationship with the d18O record, which makes us difficult to work out a good explanation. In the following research, we would expect to understand the underlying possible
mechanism by more investigation and modelling work in this region.

Minor Comments:

1) It will be better to add some information about the relationship between U-concentration and $\delta$234Ui in the results.

We will add this information and the corresponding figure in the supplementary material. By the way the correlation analysis showed that the linear relationship between 238U concentration and d234Ui are statistically insignificant (please find the plot in the supplementary file attached).

2) It is difficult to identify the difference in precision between the 234U ages and the 230Th ages from the figure 4. It will be better to provide some detailed comparisons of the two dating ages in terms of precision.

We revised this figure by deepening the color of the error bars and enlarging the inconsistent points to the level of precision. Please see the Fig. 2 attached.

3) In the conclusions, the authors should acknowledge that the conclusions are based on the regression analysis of the $\delta$234Ui, $\delta$18O and $\delta$13C datasets over the past 309 ka, but not over the past 590 ka.

We will clarify this in our conclusions.

Please also note the supplement to this comment:
https://gchron.copernicus.org/preprints/gchron-2020-26/gchron-2020-26-AC1-supplement.pdf
* * *
[Figure]

(a)

(b)

**Fig. 1.** Plots of the d234Ui, d13C and d18O curves versus the depth over the past 590 ka BP (left) and the scatter plots between d13C and d234Ui, and d18O and d234Ui with the linear regression lines (right).

**Fig. 2.** Scatter plot in the 234U ages vs 230Th ages between 4 to 590 ka BP with the corresponding 2s uncertainty. The 1:1 line and the inconsistent points (red dots) between two kinds of age are shown

**Supplement:**

Table: Comparison between multiple regression model comparison. The data with bold font refers to the model using in the MS.

| Time range (counts) | Intercept | | δ18O | | δ13C | | Residual | Statistics |
|---|---|---|---|---|---|---|---|---|
| | Value | 1σ error | Value | 1σ error | Value | 1σ error | 2σ error | Adj. $R^2$ |
| Instrumental weighting (factor=1/σ²) for 234Ui | | | | | | | | |
| 0-309 (66) | 874 | 192 | -61 | 10 | 57 | 17 | 62.4 | 0.65 |
| 0-362 (81) | 888 | 172 | -61 | 9.3 | 57 | 14 | 67.3 | 0.65 |
| 0-590 (110) | 898 | 147 | -60 | 8.0 | 58 | 12 | 78.7 | 0.65 |
| No weighting | | | | | | | | |
| **0-309 (66)** | **1219** | **189** | **-44** | **10** | **86** | **16** | **60.5** | **0.61** |
| 0-362 (81) | 1411 | 163 | -32 | 8.7 | 79 | 16 | 62.8 | 0.54 |
| 0-590 (110) | 1658 | 151 | -20 | 8.2 | 108 | 14 | 70.9 | 0.52 |

[Figure]

Figure: plots of residual generated from the regression analysis with and without weighting method over the past 309 ka BP period. The blue plot shows the difference of the residuals between two models, which is quite small relative to the variance of the residual.

[Figure]

[Figure]

Figure: scatter plots of d234Ui versus the reciprocal U concentration. The lower plot removed the outlier in the upper one, which still shows an insignificant correlation relationship.

---

## Author Comment (AC2) · 4 Nov 2020

General comments: Based on sound analytical methods the authors present an impressive data set and an innovative approach which definitely deserves publication in this journal. However, prior acceptance, I would suggest that the authors clarify some of their statistical methods and the estimation of their uncertainties. One major point is that most paleo climate time series are impacted by autocorrelation (e.g. Macias-Fauria et al. (2012);Hu et al. (2017), and others). Serial correlation is known to reduce the degrees of freedom of the time series and has to be taken into account, by, e.g. adjusting the p-value or estimating appropriate confidence intervals (Olafsdottir and Mudelsee, 2014; Zwiers and von Storch, 1995;Mudelsee, 2003). I strongly recommend that the authors discuss this issue, such as to which extent their data is affected

by autocorrelation, and how this influences their results. I would further recommend to show some evidence that the derived correlation and the regression model are not dependent on single values (such as the few data points with the lowest $\delta$234U in Fig. 1) and/or the choice of the calibration interval.

We greatly appreciate the valuable comments from the reviewers of our work. We have revised our manuscript, according to the reviewers' comments, questions, and suggestions. We believe that the manuscript has been further improved. Using the program provided in Olgfsdottir and Mudelsee (2014) with a bootstrap resample technique, we recalculate the correlation coefficients (r) between d234Ui and d18O/d13C, with the 95% confidence intervals (see the supplementary file attached), which offers more information about linear relationship between variables. The r values here are the same with that we obtained in the OriginPro software, and further confirmed that our calculation in the MS are about right but less detail description and interpretation. From the Table 1 in the supplementary file, the decreasing r between d234Ui and d18O, but with the 95% confidence intervals overlying each other to a large extent, will modify the regression model slightly, the difference between models is small with respect to the relatively large uncertainty of residual. Besides, using the Matlab-based program in Macias-Fauria et al (2012), we reconstructed the MLR models in term of the three groups split in the MS. Basically, the models are equal to the ones in the MS (see the Table 2 in the supplementary file). The R2 generated in the Matlab-based program over the period of 4-309 ka is 0.624 with p=0.025, which means the model is acceptable. Please find more information in the supplementary file attached.

Minor comments: L108ff: It is unclear which statistical method and/or settings of OriginPro have been used and how the analytical uncertainties are propagated to the predicted $\delta$234U values. OriginPro does not automatically include the uncertainties of both the y and x values in correlation and regression analyses. However, it also allows to calculate confidence as well as prediction intervals. So please clarify.

We apologize for the lack of information and we here clarify the information like this: In

the OriginPro, we choose the pairwise Pearson's correlation type to calculate the correlation coefficients with 95% confident level. For the regression analyses, user can find the "Fitting" option under the "Analysis" menu, and over there, "linear fit" and "multiple linear regression (MLR)" fitting method could be chosen for simple and multiple linear regression models, respectively, and "polynomial fit" for the simple quadratic and cubic regression. To obtain the multiple polynomial regression analysis, we firstly calculated series of square/cubic values of independent variables and then apply the MLR fitting method to establish the corresponding model. The analysis results report variety of parameters to help users to understand the model, including fit parameters (the value, standard error, t value and p value), and fit statistics (like coefficient of determination (COD), i.e. $R^2$, Adjusted $R^2$, Residual sum of squares (RSS)), analysis of variance (ANOVA), covariance and correlation matrix and residual analysis (histogram, residual lag plot and such as). For the single regression fitting method, this software also can calculate confidence and prediction bands. To ignore the analytical uncertainty of d234Ui in the regression model, we choose the part of dataset with smaller uncertainty to build the model, the model validation was discussed above and the response to the review #1. Please find the specific information in our response over there.

L132-134: In my opinion, the manuscript would be of even more value for the broader scientific community, if the main points of the proxy interpretation from the Devils Hole calcite deposits would be summarized in 1-2 more sentences. In my opinion, the description of the mechanistic understanding of the underlying processes is too short, and the authors focus mainly on the statistics. I understand that this is not the scope of the manuscript, but to support the statistical model, a proper mechanistic understanding of the underlying processes is essential. In the current version, however, one is referred to the numerous previous DH publications, which are probably not familiar to potential readers. L136-138: See previous comment. A bit more explanation of the processes would be very helpful.

We appreciate this suggestion and will extend the proxy interpretation in this paragraph as follow. DH/DH2 $\delta$18O is a reflection of meteoric precipitation at the principle recharge zones of the Ash Meadows Basin. Modern precipitation $\delta$18O varies seasonally by >10‰ in southern Nevada. Winter precipitation ($-12$ to $-14$‰ VSMOW) is sourced from the Pacific and provides the dominate fraction of aquifer recharge ($\sim$90%), while summer precipitation (0 to $-3$‰ VSMOW) is sourced from monsoonal systems from the Gulfs of Mexico and California. We interpret past variations in DH/DH2 $\delta$18O to be the result of (i) changes in temperatures and variations in the pathlength of moisture transport through Rayleigh fractionation processes, (ii) changes in $\delta$18O values at moisture source regions, and (iii) changes in the relative contributions of summer versus winter precipitation (see Mosley et al., 2016 for details). Past DH/DH2 $\delta$13C variations have been argued to reflect the extent and density of vegetation in the recharge zones of Ash Meadows Basin, such that $\delta$13C minima correspond to periods of maximum vegetation.

L145/Figure 1: Please visualize the applied linear regression model and its uncertainties

Please find the revised figure attached (Fig. 1)

L148: Compare previous comments, please state if the r and p values are corrected for autocorrelation

Here, we did not correct the r and p values for autocorrelation, and we will replace this table with Table 1 in the supplementary file.

L156-157: Again, does the adjustment of R2 already take autocorrelation into account? L168: What is the critical value of the F-test when adjusting the DF for an autocorrelated time series?

In the MS, we did not take into the autocorrelation. To estimate the effective DF from the autocorrelated time series, we use the equation veff=N * (Dt/(2*Te)) to do the rough calculation, where N is the total number of data, Dt is the average time interval between

data and Te is the persistent time. Here, we use the data over the past 309 ka. Using the program in Olgfsdottir and Mudelsee (2014) with the 230Th ages directly, we can easily obtain the persistent times for d13C, d18O, d234Ui and residual, which are 26, 15, 48 and 14ka, respectively. the ïĄĎt is about 4.7ka, and thus the effective DFs of d13C, d18O, d234Ui and residual are 6, 10, 3.2 and 11, respectively. In this case, the adjusted R2 will be 0.89, And the corrected F value is 9.2 which is still higher than the critical value of the F-test, 3.98. We will do this correction in detail in the revised MS.

L175-176: Which part of the data is treated in this part? The whole 590ka interval? Please clarify which values are used here for the validation of the regression model.

We apologize for the confusion in the part. All the residual analysis are based on the model in the MS established over the 4-309 ka interval, not the whole 590 ka interval. We will clarify this in the following revised MS.

L178-179: According to Table 4, the standard errors of the model coefficients are in the order of 15-20%, so how does the estimate of the residual uncertainty stated here compare to the uncertainty of the regression model itself? Does the width of the histogram change when taking into account the uncertainty of the regression model?

All the standard errors of the model coefficients are calculated based on the residual standard deviation, so the estimate of the residual uncertainty is basically the same with this standard errors here.

L194-195: The variability of the residuals may originate in the method used for calibration. When calibrating using linear regression, the variance of the proxy time series is always less than that of the calibration data set, since the resulting amplitude reductions are dependent on the correlation between the proxy and the calibration data set (Esper et al., 2005).

This statement indeed helps us better understand the meaning of the coefficient of determination, i.e. R2, and its relationship with the variance of residual.

L296: Reference not in alphabetical order.

We apologise for the issue and have it corrected in the MS.

Supplementary material: L20: Derived

We apologise for the typo and have it corrected.

Please also note the supplement to this comment:
https://gchron.copernicus.org/preprints/gchron-2020-26/gchron-2020-26-AC2-
supplement.pdf

**Fig. 1.** Plots of the d234Ui, d13C and d18O curves versus the depth over the past 590 ka BP (a) and the scatter plots between d13C and d234Ui, and d18O and d234Ui with the linear regression lines.

**Supplement:**

Table 1: estimated correlation coefficient with 95% calibrated confidence interval using the method in Olafsdottir and Mudelsee (2014).

| Time range | Estimated correlation (r) with 95% calibrated confidence interval | | | |
|---|---|---|---|---|
| | d18O-d234Ui | | d13C-d234Ui | |
| 0-309 ka (66) | -0.675 | [-0.948; -0.172] | 0.718 | [0.203; 0.922] |
| 0-362 ka (81) | -0.640 | [-0.904; -0.021] | 0.691 | [0.333; 0.875] |
| 0-590 ka (110) | -0.528 | [-0.810; -0.063] | 0.709 | [0.567; 0.810] |

Table 2: Regression models comparison: the weighted regression methods give about the same fit parameters regardless of the choice of the calibration interval. The regression model without weighting, is somewhat changed, presumably due to the decreased correlation between d234Ui and d18O with more data incorporated from older section (see Table 1). Data with Bold font are the model adopted in the MS.

| Time range (counts) | Intercept | | $\delta18O$ | | $\delta13C$ | | Residual | Statistics |
|---|---|---|---|---|---|---|---|---|
| | Value | $1\sigma$ error | Value | $1\sigma$ error | Value | $1\sigma$ error | $2\sigma$ error | Adj. $R^2$ |
| Instrumental weighting (factor=$1/\sigma^2$) for 234Ui | | | | | | | | |
| 0-309 (66) | 874 | 192 | -61 | 10 | 57 | 17 | 62.4 | 0.65 |
| 0-362 (81) | 888 | 172 | -61 | 9.3 | 57 | 14 | 67.3 | 0.65 |
| 0-590 (110) | 898 | 147 | -60 | 8.0 | 58 | 12 | 78.7 | 0.65 |
| No weighting | | | | | | | | |
| **0-309 (66)** | **1219** | **189** | **-44** | **10** | **86** | **16** | **60.5** | **0.61** |
| 0-362 (81) | 1411 | 163 | -32 | 8.7 | 79 | 16 | 62.8 | 0.54 |
| 0-590 (110) | 1658 | 151 | -20 | 8.2 | 108 | 14 | 70.9 | 0.52 |
| No weighting (calculated with the method in Macias-Fauria et al. (2012)) | | | | | | | | |
| 0-309 (66) | 1223 | | -44 | | 86 | | 60.5 | 0.62 |
| 0-362 (81) | 1414 | | -32 | | 80 | | 62.8 | 0.55 |
| 0-590 (110) | 1654 | | -21 | | 108 | | 70.9 | 0.54 |

Table 3: Coefficient of Determination (R2) , Reduction of Error (RE), Coefficient of Efficiency (CE), and Coefficient of Correlation (r2) with the 95% and 99% threshold values are shown

| | 4-309 ka (R2=0.624, p=0.0025) | | 4-355 ka (R2=0.553, p<0.001) | | 4-590 ka (R2=0.536, p<0.001) | |
|---|---|---|---|---|---|---|
| | Normal | Cross Cal./Ver. | Normal | Cross Cal./Ver. | Normal | Cross Cal./Ver. |
| RE | 0.235 | 0.427 | -0.037 | 0.51 | 0.351 | 0.561 |
| RE_95 | 0.435 | 0.325 | 0.248 | 0.221 | 0.267 | 0.226 |
| RE_99 | 0.605 | 0.519 | 0.361 | 0.361 | 0.396 | 0.331 |
| CE | -0.28 | 0.277 | -0.458 | 0.417 | 0.294 | 0.516 |
| CE_95 | 0.194 | 0.076 | 0.155 | 0.109 | 0.207 | 0.165 |
| CE_99 | 0.503 | 0.341 | 0.322 | 0.267 | 0.336 | 0.252 |
| R2 | 0.691 | 0.685 | 0.701 | 0.472 | 0.624 | 0.478 |
| R2_95 | 0.468 | 0.74 | 0.462 | 0.52 | 0.368 | 0.475 |
| R2_99 | 0.588 | 0.823 | 0.591 | 0.632 | 0.468 | 0.611 |
| r2 | 0.358 | 0.412 | 0.406 | 0.552 | 0.387 | 0.525 |
| f2_95 | 0.609 | 0.353 | 0.39 | 0.354 | 0.374 | 0.273 |
| r2_99 | 0.772 | 0.496 | 0.553 | 0.461 | 0.531 | 0.384 |

---

## Author Response (AR1)

[revised manuscript text omitted]

**Response to the reviewers' comments**

We appreciate all of the valuable comments from the reviewers of our work. We have revised our manuscript, according to the reviewers' comments, questions, and suggestions. We believe that the manuscript has been further improved

**Anonymous Referee #1**

*General Comments:*
*This is an interesting and excellent study that uses the correlation observed between the cave calcite δ234Ui and stable isotopes in the core from Devils Hole 2 to establish a multilinear model for prediction of the δ234Ui. This model allows the authors to predict much precisely a value of δ234Ui and thus to improve the precision of 234U-238U dating method.*

We greatly appreciate the valuable comments from the reviewers of our work. We revised our manuscript, according to the reviewers' comments, questions, and suggestions.

*Major Comments:*
*1) It lacks a figure in the MS to show the δ234Ui, δ18O and δ13C time-series.*

We modified the Figure 1 in the revised manuscript.

*2) For the regression analysis, the authors split the observed δ234Ui into three groups according to the age range and the precision. However, it will be better to find a weighting method, which can help to take into account all of the observed values of δ234Ui over the past 590 ka.*

We thank the reviewer for this helpful suggestion. We calculated the suggested regression model in the revised supplementary materials. Based on the discussion, we decided to keep the original model used in the manuscript with even greater confidence. Please find the supplementary file attached about the models and residual analysis.

*3) The figure 3 shows the variability of the residual and δ18O versus 230Th age, but its significance was poorly explained.*

We agree that although figure 3 shows an interesting observation, the significance of that observation is not yet fully understood. We have moved Figure 3 to the supplementary materials in order to avoid confusion, as it is not a main point in this study.

*Minor Comments:*

425  *1) It will be better to add some information about the relationship between U-concentration and δ234Ui in the results.*

We added a U conc vs $\delta^{234}U_i$ plot to the revised supplementary materials and added a short summary to the revised manuscript (line 71).

430  *2) It is difficult to identify the difference in precision between the 234U ages and the 230Th ages from the figure 4. It will be better to provide some detailed comparisons of the two dating ages in terms of precision.*

We modified this figure in the revised manuscript.

435  *3) In the conclusions, the authors should acknowledge that the conclusions are based on the regression analysis of the $\delta^{234}U_i$, δ18O and δ13C datasets over the past 309 ka, but not over the past 590 ka.*

We clarified this in Line 279-280 in the revised manuscript.

440  **Anonymous Referee #2**

*Li et al. present a method to extend the 230Th-234U-dated chronology of Devils Hole 2 calcite. The approach uses multi-linear regressions between calcite δ234U, δ28O and δ13C values to predict initial δ234U variability, which allows to calculate 234U-238Uages until 731ka BP with an average age precision of about 2%.*

445

**General comments:**

*Based on sound analytical methods the authors present an impressive data set and an innovative approach which definitely deserves publication in this journal. However, prior acceptance, I would suggest that the authors clarify some of their*
450  *statistical methods and the estimation of their uncertainties. One major point is that most paleo climate time series are impacted by autocorrelation (e.g. Macias-Fauria et al. (2012);Hu et al. (2017), and others). Serial correlation is known to reduce the degrees of freedom of the time series and has to be taken into account, by, e.g. adjusting the p-value or estimating appropriate confidence intervals (Olafsdottir and Mudelsee, 2014; Zwiers and von Storch, 1995;Mudelsee, 2003). I strongly recommend that the authors discuss this issue, such as to which extent their data is affected by*
455  *autocorrelation, and how this influences their results. I would further recommend to show some evidence that the derived*

*correlation and the regression model are not dependent on single values (such as the few data points with the lowest δ234U in Fig. 1) and/or the choice of the calibration interval.*

We greatly appreciate the valuable comments by the reviewers of our work. We have revised our manuscript, according to the reviewers' comments, questions, and suggestions. Using the program provided by Olgfsdottir and Mudelsee (2014) with a bootstrap resample technique, we recalculated the correlation coefficients (r) between $\delta^{234}U_i$ and $d^{18}O/d^{13}C$, with 95% confidence intervals (see the Table 2 in the revised manuscript), which offers more information about the linear relationship between variables. The r values here are the same with that we obtained in the OriginPro software. Using the Matlab-based program in Macias-Fauria et al (2012), we reconstructed the MLR models in term of the three groups split in the manuscript. Results show that the models are equal to the ones in the MS (see the Table 2 in the supplementary file). The $R^2$ obtained by the Matlab-based program over the period of 4-309 ka is 0.624 with p<0.001, which suggests that the model is acceptable. Statistical analysis using the programs above are capable of overcoming the autocorrelation issues in the climate time series. In addition, the MLR models with the weighting method shows that they are not dependent on single values. Additional details have been added to the revised supplementary file.

**Minor comments:**

*L108ff: It is unclear which statistical method and/or settings of OriginPro have been used and how the analytical uncertainties are propagated to the predicted δ234U values. OriginPro does not automatically include the uncertainties of both the y and x values in correlation and regression analyses. However, it also allows to calculate confidence as well as prediction intervals. So please clarify.*

We have added a detailed description of the OriginPro settings used to the revised supplementary file.
$\delta^{234}U_i$
We have added the following text to line 203-207 in the revised manuscript in order to clarify how analytical uncertainties are propagated to the predicted (or SD) $\delta^{234}U_i$ values: "The analytical uncertainty of $^{230}$Th-derived $\delta^{234}U_i$ is composite of (i) uncertainty of measured $\delta^{234}U_p$, which is incorporated into the $^{234}U$-$^{238}U$ ages (see the following section), and (ii) the uncertainty of $^{230}$Th ages, which is negligible relative to uncertainty of the model over the past 309 ka . Thus, we do not take analytical uncertainty into account."

*L132-134: In my opinion, the manuscript would be of even more value for the broader scientific community, if the main points of the proxy interpretation from the Devils Hole calcite deposits would be summarized in 1-2 more sentences. In my*

490 *opinion, the description of the mechanistic understanding of the underlying processes is too short, and the authors focus mainly on the statistics. I understand that this is not the scope of the manuscript, but to support the statistical model, a proper mechanistic understanding of the underlying processes is essential. In the current version, however, one is referred to the numerous previous DH publications, which are probably not familiar to potential readers.*

*L136-138: See previous comment. A bit more explanation of the processes would be very helpful.*

495  We appreciate this suggestion and expanded the proxy interpretation in Line 137-147 in the revised manuscript.

*L145/Figure 1: Please visualize the applied linear regression model and its uncertainties*

Please find the revised figure 1 in the revised manuscript.

500

*L148: Compare previous comments, please state if the r and p values are corrected for autocorrelation*

We did not correct the r and p values for autocorrelation in our original manuscript. In the revised manuscript, we recalculated the values by the method in Olgfsdottir and Mudelsee (2014). Please find the Table 2 in the revised

505  manuscript.

*L156-157: Again, does the adjustment of R2 already take autocorrelation into account?*

*L168: What is the critical value of the F-test when adjusting the DF for an autocorrelated time series?*

510  In the supplementary file we expand upon this topic.

*L175-176: Which part of the data is treated in this part? The whole 590 ka interval? Please clarify which values are used here for the validation of the regression model.*

515   It is the 309ka interval is treated in this part. We have clarified this in Line 198 in the revised manuscript.

*L178-179: According to Table 4, the standard errors of the model coefficients are in the order of 15-20%, so how does the estimate of the residual uncertainty stated here compare to the uncertainty of the regression model itself? Does the width of the histogram change when taking into account the uncertainty of the regression model?*

520

All the standard errors of the model coefficients are calculated based on the standard deviation of the residual, so the estimate of the residual uncertainty is basically the same with the standard errors here.

*L194-195: The variability of the residuals may originate in the method used for calibration. When calibrating using linear*
525 *regression, the variance of the proxy time series is always less than that of the calibration data set, since the resulting amplitude reductions are dependent on the correlation between the proxy and the calibration data set (Esper et al., 2005).*

This statement indeed helps us to better understand the meaning of the coefficient of determination, i.e. $R^2$, and its relationship with the variance of residual.
530

*L296: Reference not in alphabetical order.*

Corrected.

535 *Supplementary material: L20: Derived*

Corrected.

---

## Author Response (AR2)

[revised manuscript text omitted]

**Response to the Editors' comments (technical revision)**

**#1 comments**

**Associate Editor Decision: Publish subject to technical corrections** (24 Nov 2020) by Norbert Frank

390   Thanks for your efforts to revise the manuscript. I accept your article and recommend it for publication after some technical revision.

L162-163 J. Fiebig pers. communication - please remove as you provide a reference in addition)

    Corrected.

Table 4: correct numbering instead of 1.2E3 use 1200 (correct also p-values)

    Corrected.

395   L57 and L 204 in one you reduce uncertainty by 35% and according to your discussion this is only 40%.

    Corrected.

L 202 4.4E-13 (just remove number and note < 10-12 instead.

    Corrected.

L 278 spell out MLR in the conclusion

400     Corrected.

Please check reference list - in several places the isotope index is place behind the Isotope capital U238 etc...

    Line 338, 361 & 363: in the original papers (all published in 1960s), the titles are named in this way which places the number after the isotope capital). We would prefer to keep them intact.

Supplement Table 3:

405   Please provide data with only two significant digits. in cases 590019±30745 ->59000±31000 etc...

    We made this change in supplementary table 3.

Supplement Table 2: Remove yellow underline of first number 0.454.

    Corrected.

410 **#2 comments**

**Editor Decision: Publish subject to technical corrections (24 Nov 2020) by Klaus Mezger**

thank you for resubmitting a revised and improved version of your manuscript. It is now acceptable for publication, after some technical corrections. See list provided by the Associate Editor.

I have a few additional requests: report data properly and consistently, errors with 1-2 significant digits and the corresponding digits for

415 the data.

    We made this change in Table 4 & 5 and all the supplementary tables accordingly.

Use chemical symbols when the speciation of the element is not specified (U rather than Uranium). Use name only if you mean the material (pure metal) or at the beginning of a sentence.

    Corrected in Lines 31, 98, 105.

420 Use k for thousand and M for million, do not use the word. Right now it is all mixed in the paper for unknown reasons.

    In Line 13, 18-19, 37: we replace "thousand years" with "ka";

    In Line 45, 66, 78, 235: we replace "million years" with "Ma".

L239 values have about 1 ‰ uncertainty: should probably be: values have about 1 ‰ absolute uncertainty

Corrected.

425

**The list of all relevant changes made in the manuscript:**

**Line 13, 18, 19, 37, 45, 66, 78 and 235**: used k or M instead of thousand or million.

**Line 31, 98, 105**:  replaced "Uranium" with "U".

430   **Line 56**: corrected from 35 % to 40 %.

**Line 159**: removed "J. Fiebig pers. Communication" as citation.

**Line 177, 187 (Table 4)**: modified the value with right numbering.

**Line 198-199**: changed from "60.5" to "61" (two significant figures).

**Line 199, 205 (Figure 2b)**: substituted "$10^{-12}$" for "4.4E-13".

435   **Line 231-232 (Table 5)**: made change in the significant figures for data report.

**Line 236**: added "absolute" between "1‰" and "uncertainty".

**Line 246 (Figure 4)**: made the lines in the insert plot identical to those in the main plot and legend.

**Line 254**: replace "cf. Fig.6" with "cf. Fig.5".

**Line 275-276**: refine the sentence.